# Surface charge-induced orientation of interfacial water suppresses heterogeneous ice nucleation on α-alumina (0001)

Ahmed Abdelmonem[1], Ellen H. G. Backus[2], Nadine Hoffmann[1], M. Alejandra Sánchez[2], Jenée D. Cyran[2], Alexei Kiselev[1] and Mischa Bonn[2]

[1]Institute of Meteorology and Climate Research – Atmospheric Aerosol Research (IMKAAF), Karlsruhe Institute of Technology (KIT), 76344 Eggenstein-Leopoldshafen, Germany
[2]Max Planck Institute for Polymer Research, Ackermannweg 10, 55128 Mainz, Germany

*Correspondence to:* Ahmed Abdelmonem (ahmed.abdelmonem@kit.edu)

**Abstract.** Surface charge is one of the surface properties of atmospheric aerosols, which has been linked to heterogeneous ice-nucleation and hence cloud formation, microphysics and optical properties. Despite the importance of surface charge for ice nucleation, many questions remain on the molecular-level mechanisms at work. Here, we combine droplet freezing assay studies with vibrational sum frequency generation (SFG) spectroscopy to correlate interfacial water structure to surface nucleation strength. We study immersion freezing of aqueous solutions of various pHs on the atmospherically relevant aluminum oxide α-$Al_2O_3$ (0001) surface using an isolated droplet on the surface. The high pH solutions freeze at temperatures higher than that of the low pH solution while the neutral pH has the highest freezing temperature. On the molecular level, the SFG spectrum of the interfacial water changes substantially upon freezing. At all pHs, crystallization leads to a reduction of intensity of the 3400 cm$^{-1}$ water resonance, while the 3200 cm$^{-1}$ intensity drops for low pH but increases for neutral and high pHs. We find that charge-induced surface templating suppresses nucleation, irrespective of the sign of the surface charge. Heterogeneous nucleation is most efficient for the nominally neutral surface.

## 1 Introduction

The optical properties of the atmosphere play a key role in determining the climate through a combination of reflected incident solar radiation and trapping of outgoing infrared radiation. These processes occur in the clouds and are strongly influenced by the formation of ice particles in clouds (Baker and Peter, 2008), which can occur in two manners: either homogeneously, typically at temperatures below -38 °C or, when formed above this temperature, through a heterogeneous nucleation process triggered by aerosol particles present in clouds (Pruppacher and Klett, 1997). The heterogeneous freezing of supercooled water on an ice nucleating agent constitutes the main mechanism of ice particle formation in the atmosphere (Baker and Peter, 2008; Murray et al., 2012; Hoose and Mohler, 2012; Ladino Moreno et al., 2013). The nucleation rate depends on surface area, roughness and charge of the aerosol particle, as well as on the temperature and relative humidity. It is suggested that ice nucleation occurs preferentially at microscopic active sites and is dominated by the nature of these sites rather than by the average behavior of the surface (Bryant et al., 1960; Corrin and Nelson, 1968; Zettlemoyer et al., 1963; Kiselev et al., 2016).

Mineral dust can initiate ice formation at low saturations and temperatures warmer than homogeneous freezing and, thus, influence ice cloud properties (Archuleta et al., 2005; Kanji and Abbatt, 2006; Möhler et al., 2006). Kulkarni and Dobbie studied the heterogeneous ice nucleation properties of three Saharan (from Nigeria, outside of Dakar city, and Dakar) and one Spanish (South East coast) dust particle samples and attributed the wide scatter in ice nucleation efficiency data to two reasons: First,

different surface elemental compositions and second, variation of surface irregularities or roughness across the dust particles (Kulkarni and Dobbie, 2010). Elemental analysis of their dust particles across the four source regions revealed variations in elemental composition with a majority of Si, Al, Mg, Ca, Na, and Fe and minority (<1%) of other elements, including P, K, Ti, and Cl. minerals. These variations of elemental composition might be responsible for variations of the ice nucleation efficiencies.

The closest study of ice nucleation on minerals relevant to the atmospheric dust particles mentioned above in terms of elemental compositions was provided by Eastwood et al. (Eastwood et al., 2008). They found that dust minerals of different elemental composition have a wide range of onset nucleation supersaturation with respect to ice. Minerals, such as quartz and calcite were observed to be poor ice nucleators, while muscovite, kaolinite, and montmorillonite exhibited higher ice nucleation ability. Recently, feldspars were identified to be among the most active atmospherically relevant ice-nucleating minerals (Atkinson et

al., 2013; Harrison et al., 2016). The ice nucleation properties of several alkali -feldspars were studied recently using the droplet freezing assay setup (Peckhaus et al., 2016). It was found that K-rich feldspars (microcline) exhibit a high ice-nucleating efficiency, whereas Na/Ca-rich feldspar freezes at a lower temperature. However, the real influence of different ions on the freezing process remains ambiguous. The real influence of temperature and supersaturation on the interaction between water molecules and dust particle surface has been the focus of research since decades but remains debated. Cloud pH may change

depending on the concentration of acidic or alkaline particulates in atmosphere. It is has been shown that the acidic particulates (e.g. $SO_4$, and $NO_3$) from anthropogenic sources decrease the pH values in cloud and rain water (Castillo et al., 1983; Scott, 1978). On the other hand, it was reported that alkaline particulates were observed in the regions where soil is rich with alkaline components, e.g. Ca and Mg, (Khemani et al., 1985b). In these regions high pH values were observed in cloud and rain water (Khemani et al., 1985a; Khemani et al., 1987). The aerosol particle itself can have acidic or alkaline components absorbed on its

surface which may dissolve in the water droplet and change its pH. The water droplet size affects the solute concentration (Noone et al., 1988) and hence pH. Since immersion freezing is based on aerosol particles immersed in water droplets in a cloud for a certain time, their surface charge may change due to the variation in the droplet pH because the surface charge of a metal oxide surface is pH dependent (Kosmulski, 2001, 2009). The higher the pH with respect to a specific pH, known as the point of zero charge (pzc) for which the surface is nominally neutral, the more negative is the surface, while the lower the pH with

respect to the pzc the more positive is the surface. Surface charge is one of the surface properties which influence its interaction with water molecules, hence we believe that investigating the freezing-pH dependence will help in understanding one of the not well explored parameters of ice nucleation in the atmosphere. In addition, we found in a former study that the re-adsorption of dissolved ions on the surface of mineral oxide in aquatic environment can change the surface charge (Lützenkirchen et al., 2014). We expect that this change in the surface charge is to affect the ice nucleation ability of the surface.

Despite the surface of mineral aerosol particles playing an important role in heterogeneous nucleation, the details of the different aspects of the surface (charge, corrugation, etc.) have remained poorly understood. Recent studies performed on polar organic crystals have demonstrated that surface charge can have a strong effect on heterogeneous freezing of supercooled water, but the suggested mechanism has not been conclusively verified (Belitzky et al., 2016; Ehre et al., 2010; Gavish et al., 1992).

Yang et al. have previously suggested, from vibrational SFG spectroscopy at the water-mica interface for solutions with different

molarities of sulfuric acid at room temperature, that structured water at the interface may be required for efficient heterogeneous ice nucleation (Yang et al., 2011). Yang et al. based this conclusion on the decrease in ordered water structure, at room temperature, and the corresponding reduced ice nucleation efficiency with the increase of sulfuric acid concentration in solutions in contact with the surface. In recent studies of water in contact with sapphire by Anim-Danso and coworkers an increase in the SFG signal following the ice formation for low pH solutions (positively charged surface) and a decrease in the SFG signal at

high pH (negatively charged surface) was observed (Anim-Danso et al., 2013). The decrease of signal at high pH was attributed

to the probable segregation of sodium ions next to the negatively charged sapphire substrate, which may disrupt the charge transfer and stitching bilayer. From a similar attenuation of the SFG signal upon freezing for three different bases the cation specific effect was ruled out (Anim-Danso et al., 2016). Moreover, they concluded that the orientation of water next to the surface is a determining factor in the ice formation. Although they observed different changes in water structure upon freezing for different pH solutions, the freezing transition temperatures was found to be independent of the surface charge. However, the observed freezing temperatures was between -5 to -7 °C, which is remarkably high considering sapphire is a poor ice nucleator (Yakobi-Hancock et al., 2013; Richardson, 2006; Thomas, 2009). Abdelmonem et al. (Abdelmonem et al., 2015), using optical second harmonic generation (SHG) to probe ice nucleation on the surface of sapphire (0001), observed a freezing temperature about -15 °C, which is still high for sapphire. However, ice nucleation was concluded to occur at the interface between the crystal surface and the cell rather than at the sapphire water interface in these measurements (Abdelmonem et al., 2015). Similar effects could be the reason for the high freezing temperature reported in (Anim-Danso et al., 2016; Anim-Danso et al., 2013). To avoid early freezing which may be triggered by a different surface, as described above, we ensure in the current work that we study specifically heterogeneous nucleation at the sapphire-water interface, by isolating water drops on the surface using silicon oil. Studying immersion freezing on isolated drops on a surface has previously been reported by several groups (Broadley et al., 2012; Hama and Itoo, 1956; Murray et al., 2011; Peckhaus et al., 2016). However, this is the first time where the freezing of an isolated drop on the surface is probed by SFG spectroscopy.

SFG is a powerful surface-sensitive spectroscopic technique allowing to probe the different species at an interface, through their vibrational bands, which makes this technique useful to extract chemical and structural information of different molecules near surfaces and interfaces (Du et al., 1994; Du et al., 1993; Hsu and Dhinojwala, 2012; Jena et al., 2011; Ji et al., 2007; Rangwalla et al., 2004; Richmond, 2001; Richmond, 2002; Shen and Ostroverkhov, 2006; Yeganeh et al., 1999; Gragson and Richmond, 1998). In the dipole approximation, SFG, which is a second-order nonlinear technique, is only active where there is a breakdown in inversion symmetry. This selection rule makes it possible to use SFG to study the structure of molecules between two isotropic media. When the field of an intense visible light of frequency $\omega_v$ spatially and temporally overlaps with that of an IR tunable light of frequency $\omega_{IR}$ at the interface, a third field is generated with a frequency equals to the sum of both ($\omega_{SF} = \omega_v + \omega_{IR}$). The generated field has an intensity given by:

$$S(\omega_{SF}) \propto \left| \left[ \vec{L}(\omega_{SF}) \cdot \hat{e} \right] \cdot \overleftrightarrow{\chi}_s^{(2)} : \left[ \vec{L}(\omega_v) \cdot \hat{e} \right] \left[ \vec{L}(\omega_{IR}) \cdot \hat{e} \right] \right|^2 I_1 I_2 \tag{1}$$

where, $\hat{e}$ is the unit polarization vector, $\vec{L}(\omega_i)$ and is the tensorial Fresnel coefficient of the surface at $\omega_i$ and $\overleftrightarrow{\chi}_s^{(2)}$ is the surface nonlinear susceptibility tensor. To extract information about the different vibrational bands, the spectra can be fitted with a sum of a nonresonant contribution and a number of lorentzian line shapes:

$$\overleftrightarrow{\chi}_s^{(2)} = \overleftrightarrow{\chi}_{NR}^{(2)} + \sum_q \frac{\overleftrightarrow{A}_q}{\omega_{IR} - \omega_q + i\Gamma_q} \tag{2}$$

where $\overleftrightarrow{A}_q$, $\omega_q$, and $\Gamma_q$ are the amplitude, frequency, and damping factor of the $q^{th}$ vibrational resonance, respectively. $\overleftrightarrow{\chi}_{NR}^{(2)}$ is the nonresonant contribution. The obtained spectra can be fit by eq (2) to deduce $\overleftrightarrow{A}_q$, $\omega_q$, and $\Gamma_q$ for each resonance. Below, we will use the ratio $\overleftrightarrow{A}_q / 2\Gamma_q$ as a measure of the number density of oriented surface molecular groups.

Here, we investigate the effect of surface charge on the immersion freezing of water. As a model system, we use aluminum oxide (α-Al$_2$O$_3$), also known as sapphire or corundum, which is an atmospherically relevant oxide surface as reported in (Kanji and Abbatt, 2006; Al-Abadleh and Grassian, 2003; Brownlee et al., 1976). In addition to ion ad/de-sorption (Lützenkirchen et al.,

2014), the net surface charge of metal oxide surfaces can be tuned by the protonation or deprotonation of hydroxyl groups, which terminate the surface (Covert and Hore, 2016; Geiger, 2009). The pzc for $\alpha$-$Al_2O_3$ (0001) single crystal has been reported to be between pH= 5.3 and 7.3 (Fitts et al., 2005; Franks and Meagher, 2003; Kershner et al., 2004; López Valdivieso et al., 2006; Veeramasuneni et al., 1996; Zhang et al., 2008). We study, using SFG spectroscopy, the structure of the water molecules at the interface with $\alpha$-$Al_2O_3$ (0001) before and after freezing. The effect of surface charge on the ice nucleation was studied by varying the bulk pH of the aqueous solution in contact with the surface. This work provides molecular level information about the influence of the surface charge on the heterogeneous ice nucleation process in terms of the structure of water molecules and onset temperature. Specifically, we demonstrate that surface charge reduces the nucleation ability of the sapphire surface, irrespective of the sign of that charge.

## 2 Experimental

To investigate the effects of surface charge on the freezing temperature, independent droplet freezing assays using an equipped cold-stage were carried out to determine the exact freezing point of each solution on the sapphire (0001) surface. A detailed description of the method can be found in (Peckhaus et al., 2016). Briefly, about 500 droplets with a volume around 0.2 nL of the sample solution were printed in a regular array on the $\alpha$-$Al_2O_3$ (0001) substrate using a piezo-driven drop-on-demand generator (GeSIM, Model A010-006 SPIP, cylindrical case). The substrate was pre-cooled to the ambient dew point to reduce the evaporation of droplets. After printing, the droplet array was covered with silicone oil (VWR, Rhodorsil 47 V 1000) to prevent evaporation and any possible interaction between the supercooled and frozen droplets. Subsequently, the substrate was cooled down with a constant rate. The fraction of frozen droplets ($f_{ice}$) as function of temperature allows for determination of the median freezing temperature, which is the temperature at which half of the droplets freeze.

In order to study the heterogeneous freezing at the sapphire-water interface at the molecular level, a custom-designed sample cell was used in the SFG experiments. In the SFG experiments femtosecond infrared (IR) and spectrally narrowed visible (VIS) pulses are mixed at a sapphire prism-water interface in a co-propagating, total internal reflection geometry from the prism side to generate the SFG light, Fig. 1. The reflected SFG light is spectrally dispersed by a monochromator and detected by an electron-multiplied charge coupled device (EMCCD, Andor Technologies), see Fig. 1 upper panel. Typical liquid and ice spectra are shown in the lower panel right. In our supercooled SFG experiments, the sapphire prism is place in a cupper adaptor which is fixed on the silver block, the cooling/heating element, of the Linkam cold-stage. The cold-stage can perform controlled heating and cooling ramps, applied to the silver block, at rates between 0.01 and 100 °C/min. Temperature stability of the cold-stage is better than 0.1 K. Detailed description and drawing of the assembly of the SFG measuring cell can be found in (Abdelmonem, 2017). We cool the sample stepwise at a rate of 1 °C/min and a step size of 1 °C. At each integer of degree the temperature hold constant for one and half minute and then a spectrum is collected. The acquisition time per spectrum is 30 sec. The spectra change slightly and gradually with cooling (primarily due to temperature-dependent optical constants). At the transition point, a significant change in the signal is observed and, simultaneously, visual inspection reveals that the droplet is frozen. Figure 1, lower panel left, shows images of typical droplets on the substrate before and after freezing during the supercooled SFG experiments.  The spectra of liquid and ice we discuss here are those collected right before (liquid) and immediately after (ice) the freezing of the droplet, respectively: a collected spectrum is thus either of pure liquid or pure ice. The measuring cell, described in previous work (Abdelmonem et al., 2015), is made of Teflon and has dimensions of 15 x 15 x 50 $mm^3$ with a circular opening of 8 mm diameter on one side at its lowermost part. In this work, the cell is equipped with a stainless steel cover with three internal tubes. The first tube is used to place the water droplet on the substrate surface, the second tube to fill the cell

with silicon oil and the last tube for equilibrating the internal pressure of the cell with ambient pressure. The substrate of interest was sealed to the circular opening and the cell was filled with silicon oil, only up to the level covering the circular opening. The prism was secured by an adaptor made of copper and was in direct contact with the cooling/heating element of the cold-stage. The oil covering the droplet solution at the surface (Fig. 1) prevents a droplet from early freezing and initiation of freezing at the edge between the cell and the prism,  an effect that was observed in (Abdelmonem et al., 2015). The silicon oil also prevents the partial evaporation of the droplet and re-condensation on the rest of the cold sample surface during cooling. The water droplet had a diameter around 4 mm. Assuming that the droplet is a hemisphere, the volume of the droplet is roughly 15 µL. However, controlling the size of the droplet was difficult. This affected our accuracy in comparing the freezing temperatures of the different solutions, due to nonequivalent coverage of surface sites, and hence no statistics were obtained from the SFG measurements. In the SFG experiments, the freezing temperatures of different solutions ranged from -19 to -27 °C. The heterogeneous nucleation temperatures reported in the "Results and Discussion" section were obtained from the droplet freezing assay results shown, e.g., in Fig. 2a.

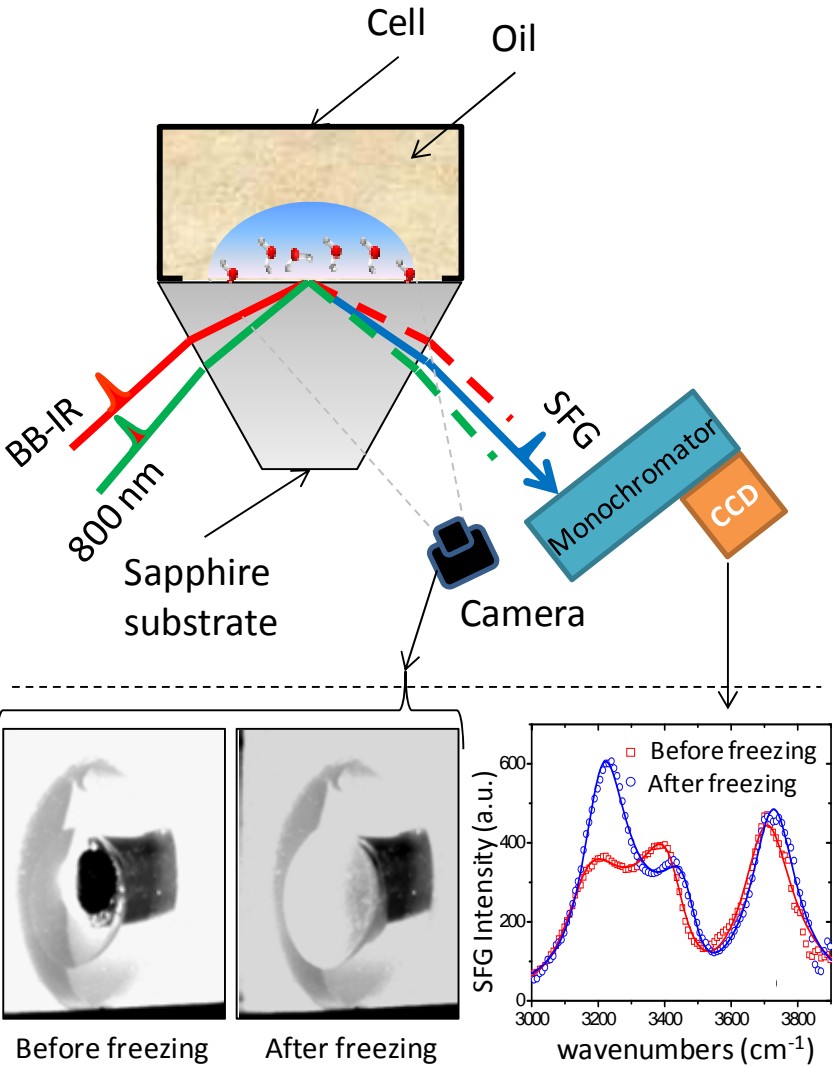

**Figure 1. Upper panel: Schematic of the sample setup used in the SFG experiments. The SFG signal is generated in a co-propagating total internal reflection geometry at the spatial and temporal overlap of the incoming visible (800nm) and infrared (BB-IR) light, focused down to a ~100 µm diameter spot size. The camera (Guppy F-036 Allied Vision Technology with LINOS Macro-CCD Lens 0.14x (1:7) f4) is used to observe the droplet while placing it on the surface and filling the cell with oil, and to observe its status during the experiment. Lower panel left: images of a typical drop on the substrate before and after freezing. Lower panel right: Typical water spectra before and after the freezing.**

For consistency, the data reported in this work were all collected on the same sapphire prism (Victor Kyburz AG, Safnern, Switzerland, roughness: 1.5 nm, flatness: lambda/4) with its basal plane exposed to the solution. The new prism sample was cleaned according to the recipe published in (Rabung et al., 2004) which reported that this cleaning procedure eliminates the organic carbon contamination and minimizes inorganic contamination (Lützenkirchen et al., 2010). In addition, we do not see CH signal from organic carbon contamination in our SFG spectra. The fresh sample was soaked in acetone overnight, subsequently washed with ethanol, then soaked in ethanol (2 h), washed with MilliQ water (18.2 MΩ.cm) and finally soaked in MilliQ water (1 h). After each experiment, the sample was soaked (~2 h) in, and then washed with, chloroform, acetone, and ethanol, respectively. As a last step before each experiment, the sample was rinsed thoroughly with MilliQ water. The contact angle of MilliQ water was measured by the drop method to be 30° for the cleaned surface.

The different pH solutions were freshly prepared in our lab and directly used in the experiment. The high pH solutions were prepared from NaOH (Sigma Aldrich) and $NH_3$ (28 % Spectrum Chemical), while the low pH solution was prepared from $HNO_3$ (65 % Sigma Aldrich). All solutions were made by diluting the chemicals in MilliQ water until the desired bulk pH, measured at room temperature using a Toledo MP 225 pH meter. Please note that the pH value is temperature dependent as a result of changes in the dissociation constant with temperature, particularly for high pH in which case the pH increases with decreasing temperature (Bandura and Lvov, 2006; Zumdahl, 1993). This will however not affect our interpretations in terms of low, neutral and high pH. In addition, the pH is only tuned to change the surface charge, which is determined independently from the water alignment using SFG. Hence, knowledge of the precise temperature dependence of the pH is not required to correlate surface charge to freezing temperature. Changes in the amount of dissolved gases in the liquid solutions were assumed to be minimal since the measuring cell was always closed.

## 3 Results and Discussion

Droplet freezing assay experiments were performed on the sapphire (0001) interface in contact with solutions of different pH. Given the pzc being between pH = 5.3 and 7.3 as mentioned above, we used solutions of pH 3.1 ($HNO_3$), pH 4.1 ($HNO_3$), pH 7.2, pH 8.3 (NaOH) and pH 8.5 ($NH_4OH$), to make sure that we investigated ice nucleation at negative (high pH), positive (low pH) and (near-) neutral charge (neutral pH). As shown in the supporting information, the small differences in ionic strength do not affect the nucleation process. To determine the freezing temperature, about 500 droplets of each pH solution were placed on a sapphire (0001) substrate and covered with silicon oil. Subsequently, the system was cooled down with a constant rate of 5° C/min. Figure 2a shows the fraction of frozen droplets $f_{ice}$ as a function of temperature for each pH solution. Figure 2b summarizes the median freezing point as function of pH. Figure 2c shows images of typical droplets on the substrate before any, at half, and after all of droplets frozen, from up to down respectively. The freezing temperatures on sapphire 0001 surface are -31.4, -30.6, -29.2, -30.6 and -29.9 °C for pH 3.1, pH 4.1, pH 7.2, pH 8.3 and pH 8.5, respectively. To ensure that the water-oil interface is not acting as a nucleation site in our experiments, all pH solutions were tested on a silicon wafer with droplets covered by silicon oil. If nucleation occurs at the water-oil interface, the freezing temperature for the sapphire and silicon substrates should be the same. However, the measured freezing temperature on the silicon wafer was -36.1 °C ±0.2 °C for all pH solutions (see Fig. 2a for a typical curve), which is close to the homogeneous freezing point and significantly lower than on the sapphire substrate. Thus, we are sure to study the nucleation directly at the sapphire-water interface. The median freezing temperature measurements for the pH solutions on a Si substrate allowed us estimate the water activity in the solutions. Applying

Hildebrand and Scott equation (Miyawaki et al., 1997) and assuming temperature measurements accuracy of ± 0.2 K the water activity has been calculated to be within the range of 0.993 to 0.996. This confirms our statement that the solute effect on variation of the freezing temperature is negligible.

The droplet freezing assays on sapphire reveal a maximum in freezing temperature, i.e. the most efficient interfacial nucleation of ice, near neutral pH. This prompts the question: how does the neutral surface enhance the ice nucleation process? Answering this question requires understanding the configuration of water molecules in the interfacial region. Therefore, we investigated interfacial water while freezing occurs, right next to the sapphire interface with SFG spectroscopy.

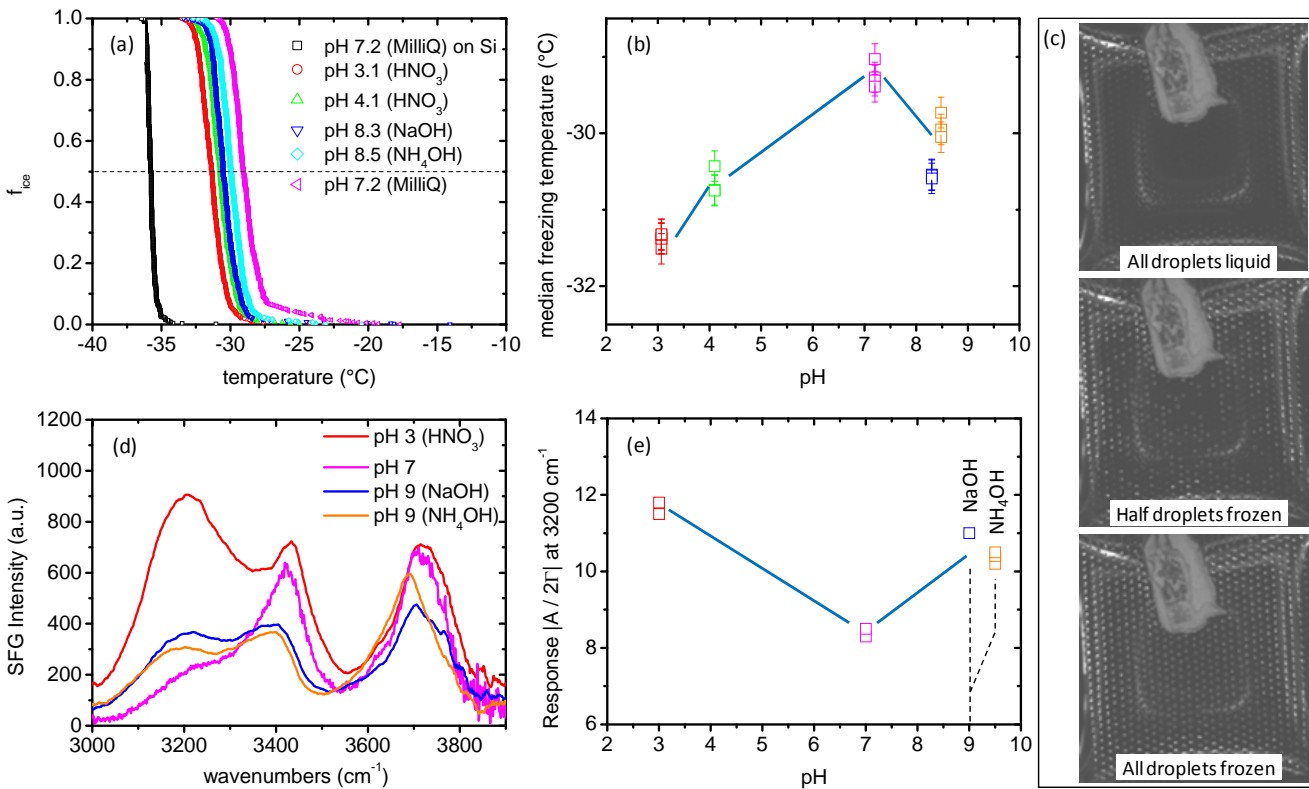

**Figure 2. (a) Fraction of frozen droplets as a function of temperature of Sapphire (0001) surface in contact with different pH solutions. The black curve shows the data for water on a silicon substrate. (b) Freezing temperature as a function of pH of the different solutions for three independent experiments. (c) Images of typical droplets on the substrate at different stages of the droplet freezing assay experiment. (d) SFG spectra of liquid water at the Sapphire (0001) surface, right before the nucleation event (see text for details); for fits, see Fig. 3. (e) Absolute value of the amplitude divided by width as a function of pH for the 3200 cm$^{-1}$ resonance inferred from the data shown in (d). Different data points indicate data obtained from freshly prepared surfaces, recorded on different runs.**

Figure 3 shows the Fresnel factor-corrected (see SI for details) SFG intensities under SSP polarization combination (s-polarized SFG and VIS; p-polarized IR) for the liquid (red) and ice (blue) phase measured for the pH 3 (HNO$_3$), pH 7, pH 9 (NaOH) and pH 9 (NH$_4$OH) solutions in contact with the sapphire (0001) surface. All spectra contain three peaks around 3200, 3400, and 3700 cm$^{-1}$, however with different relative intensities for the different samples. The peak at ∼3700 cm$^{-1}$ has been assigned to surface hydroxyl (Al$_2$OH) groups with Al octahedrally bonded and OH protruding from the (0001) surface of sapphire (Sung et al., 2012; Zhang et al., 2008; Liu et al., 2005). The signal at 3200 and 3400 cm$^{-1}$ originates from hydrogen bonded water with strong and weak hydrogen bonding interactions, respectively (Rey et al., 2002). For the water-air interface it is well-known that the broad band is additionally split into two due to inter- and intramolecular coupling (Schaefer et al., 2016). For the water-

sapphire interface this coupling might play a role as well, but it is not the dominant effect, as Shen and co-workers (Zhang et al., 2008) have reported phase resolved SFG experiments showing that the signal at low and high frequency at high pH have opposite sign. Irrespective of the precise origin, the broad band can be phenomenologically decomposed into peaks at 3200 and 3400 cm$^{-1}$ representing strongly and weakly hydrogen-bonded groups. The spectra of the supercooled liquid state are largely in agreement with spectra reported for the water-sapphire (0001) interface at room temperature (Zhang et al., 2008). For the pH 9, NH$_4$OH, sample, the N-H symmetric stretch (at ~3300 cm$^{-1}$) and overtone of the anti-symmetric angle deformation (at ~3200 cm$^{-1}$) modes might overlap with the water vibrations (Simonelli et al., 1998; Simonelli and Shultz, 2000). However there is no need to invoke a contribution from N-H groups to account for the experimental data, which is expected, given the relatively low concentrations we use here.

The SFG spectrum and the interfacial water structure change substantially upon freezing: the high frequency signal remains more or less constant in all cases, while the low frequency peaks change. This is in agreement with the designation above, assigning the 3200 and 3400 cm$^{-1}$ resonances to O-H stretch vibrations of water, which are expected to change upon water crystallization, and the one at 3700 cm$^{-1}$ to sapphire-bound OH groups, which one indeed would expect to remain unchanged upon crystallization. Regarding the water resonances, at all pHs, crystallization leads to a reduction of intensity of the 3400 cm$^{-1}$ resonance, while the 3200 cm$^{-1}$ intensity signals drops for pH 3, but increases for pH 7 and 9.

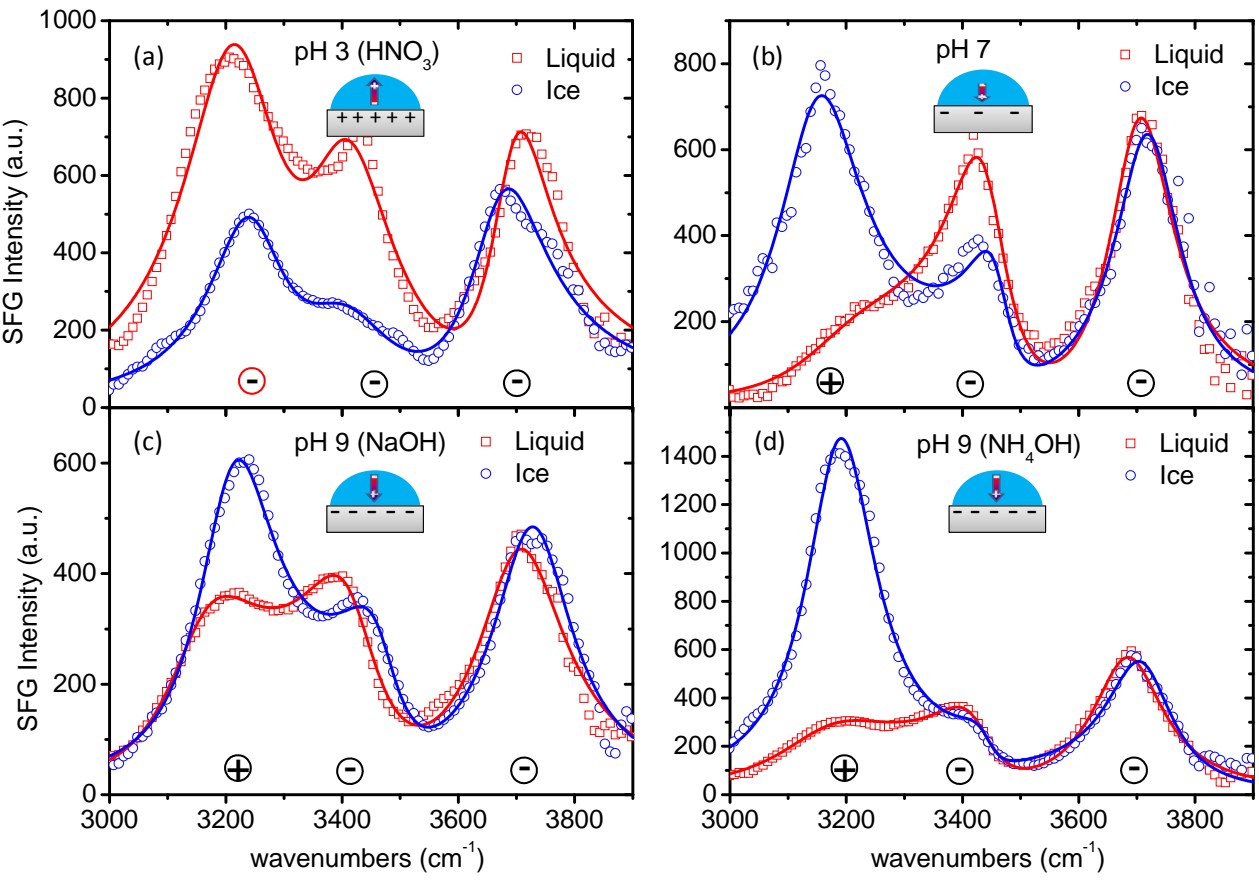

**Figure 3. Measured (points) and fitted (lines) SFG spectra of α-Al$_2$O$_3$ (0001)/water interfaces collected in SSP polarization during the cooling cycles of (a) pH 3 - HNO$_3$, (b) pH 7, (c) pH 9 - NaOH and (d) pH 9 - NH$_4$OH. The spectra shown here are always the last and first spectrum right before (red) and immediately after (blue) the freezing event, respectively. The signs of the resonances are indicated by a circle with + or - sign (see SI for details). The insets in a, b, c and d illustrate the net dipole orientations (from O to H, as inferred from the low frequency peak) of water on the surface at different pHs (surface charges).**

The reduction in the 3400 cm⁻¹ intensity upon crystallization observed at all pHs is unsurprising, as the ice response is simply redshifted from the water response due to the stronger inter- and intramolecular coupling in ice. Along the same lines, one would expect the 3200 cm⁻¹ intensity to increase upon crystallization. However, as is clear from Fig. 3a the intensity around 3200 cm⁻¹ decreases upon freezing. This absence of increase indicates that the observations are not trivial. It demonstrates that freezing-induced reorientation contributes to the changes in the 3200 cm⁻¹ response.

The spectral changes can be quantified by modeling the SFG spectra using a sum of three Lorentzian bands for the resonant contributions as described above. The signs of the amplitude of the different bands are taken from previous phase-resolved SFG experiments of Zhang et al (Zhang et al., 2008) and are assumed to not change upon freezing. The 3400 cm⁻¹ and 3700 cm⁻¹ band have a negative sign in all cases, while the 3200 cm⁻¹ band is negative for low pH (red circle in Fig. 3a), but positive for neutral and high pH. In the analyses, the nonresonant amplitude and phase are fixed. The parameters of the Lorentzians are varied to obtain the best description of the data, which are presented as lines in Fig. 3. Figure 4 depicts the amplitude divided by the width of each resonance ($\vec{A}_q/2\Gamma_q$). This variable represents the strength of the oscillator normalized to the homogeneous broadening of the band and thus is a measure for the number density of the different surface groups (Backus et al., 2012; Li et al., 2015; Pranzetti et al., 2014). Figure 4a shows that the sign and strength of the 3700 cm⁻¹ peak is largely insensitive to pH and to whether the water is in the liquid or solid state. This is consistent with this peak originating from surface-bound OH groups and that the H atoms are strongly bonded to the surface oxygen, not likely to participate in H-bonding with adsorbed water molecules and not easily deprotonated. The band at ~3400 cm⁻¹ is pH dependent in the supercooled liquid phase (red circles, Fig. 4b) which is similar to the behavior observed at room temperature (Zhang et al., 2008). The changes in the 3400 cm⁻¹ band upon crystallization are very similar for all pHs, and can be assigned to the redshifted response of ice, as mentioned above.

The signal strength at 3200 cm⁻¹ in the supercooled liquid phase (Fig. 4b) shows a pH dependence very similar to that observed for this surface at room temperature (Zhang et al., 2008): the signal intensity is lowest close to the pzc and high at low and high pH, with opposite sign at the two extremes. The negative sign of the 3200 cm⁻¹ band at pH 3 shows that water molecules with this frequency have OH groups pointing with its H into the bulk solution, inset Fig.3a.

The reduction of the 3200 cm⁻¹ signal upon freezing at pH=3 is an indication of a competition between the surface charge, aligning the OH groups toward the bulk, and the tendency of molecules to fit into the ice structure which is, apparently, requiring the OH groups to rotate, on average, towards the surface. Having the lowest freezing point amongst the four solutions shows that this competition hinders freezing. Inversely, at pH 7 and 9 the amplitudes of the 3200 cm⁻¹ signal increases upon freezing, i.e. the ice signal (blue squares in Fig. 4b) is larger than the water signal (red squares in Fig. 4b). This indicates that also here, ice prefers an orientation with the OH groups pointing towards the surface, inset Fig. 3b, c and d. The net orientation with OH pointing with its hydrogen to the sapphire already in the liquid case resembles the preferred structure for ice. One might expect this templating to facilitate the freezing process, yet the freezing temperatures at high pH are not that different from those observed at low pH. In fact, the highest freezing temperature (i.e. easiest nucleation) is observed for near-neutral pH, where the surface charge is minimal, inset Fig. 3b. This implies that any form of interfacial templating of water reduces the probability of nucleation for this sapphire surface: neutral water has the highest freezing temperature (-29.2 °C at pH 7.2) and the best ice nucleation condition lies near the pzc.

To summarize this conclusion, we plot in Fig. 2d the SFG spectra for the different liquid solutions to illustrate the pH dependent changes. Moreover, Fig. 2e depicts the absolute amplitude associated with the 3200 cm⁻¹ peak as a function of pH right before freezing. This amplitude is a direct measure for the alignment of the water in front of the Sapphire surface. As expected, there is a minimum in the degree of alignment at near-neutral pH, close to the pzc. The minimum alignment corresponds to the maximum freezing temperature. The water molecules at pH close to the pzc are loosely attached to the surface and suffer less

binding forces than those at low and high pH, and hence are more prone to nucleate. Strong H-bonding and surface field-induced alignment, at low and high pH, may restrict the freedom of the interfacial molecules to structure themselves in the ice crystal. The similar behavior for NaOH and NH₄OH indicates that the nucleation process does not depend on ion specific effects.

In terms of thermodynamics, a charged surface orients OH dipoles of the interfacial water molecules parallel to one another and therefore reduces the entropy and raises the free energy of the growing ice embryo (Fletcher, 1959; Marcolli et al., 2016). This explains the reduction of the freezing efficiencies for charged interfaces by a charge-induced surface templating, which impedes ice nucleation, irrespective of the sign of the surface charge. Interestingly, whereas sapphire is a poor ice nucleator, the enhancement of the ice nucleation ability close to the pzc has been observed previously for silver iodide, which is considered an excellent ice nucleation surface (Edwards and Evans, 1962; Marcolli et al., 2016).

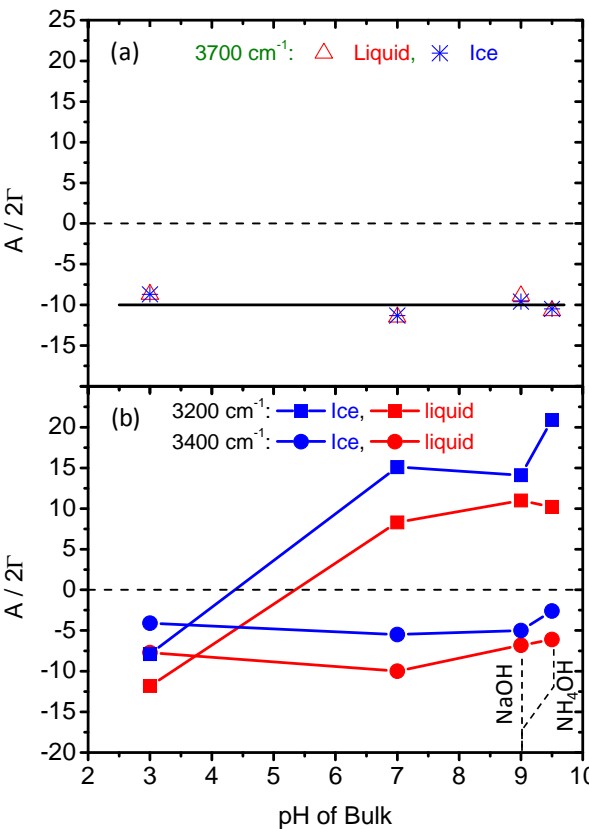

**Figure 4. Amplitudes of indicated resonances divided by their width as a function of pH for (a) the 3700 cm⁻¹ resonance and (b) the 3200 and 3400 cm⁻¹ resonances, all three inferred from Lorentzian fits to the spectra of Fig. 3. Low and high pH, may restrict the freedom of the interfacial molecules to structure themselves in the ice crystal.**

**Conclusions**

The effect of surface charge on heterogeneous ice-nucleation ability of α-alumina (0001) surface has been studied by combining freezing assays and SFG characterizations. The droplet freezing assay measurements allowed us to quantify the median freezing temperatures of the different solutions, while the SFG allowed us to probe the rearrangement of the water molecules at the interface on the molecular level. The use of an isolated water drop instead of a bulk solution in the SFG experiments ensured reaching the real heterogeneous nucleation point of the surface. Aluminum oxide can be used as a model surface of mineral aerosols and was reported as an atmospherically relevant aerosol. To investigate the effect of surface charge, which changes with the acidity or basicity of the cloud, on the immersion freezing, we studied the freezing of solution droplets of different pHs on the surface of α-alumina (0001). The selected range of pHs allowed us to study positive, neutral, and negative surfaces. The high

pH solutions (positively charged surface) were frozen at temperatures higher than those of the low pH solutions (negatively charged surface) while the moderate pH (neutral surface) had the highest temperature of freezing defining optimum ice-nucleation conditions. The SFG spectra revealed substantial changes in the structure of the interfacial water upon freezing. Low pH solutions showed disordering while at moderate and high pH freezing yielded preferential orientation of water molecules. We found that water in contact with the α-alumina (0001) surface freezes most readily when the interfacial water molecules are not well-ordered which indicates that charge-induced surface templating is detrimental for ice nucleation on this particular surface, regardless of the sign of the surface charge. Understanding the role of charge in the ice-nucleation efficiency of metal oxides is important for heterogeneous ice nucleation in atmosphere, but may also impact other environmental and industrial applications.

**Supplement**

Supporting Information Available. The supporting information comprises details of the SFG experiments and the data analysis. This material is available free of charge via the Internet at

**Acknowledgements**

AA is grateful to the German Research Foundation for support (DFG, AB 604/1-1). EHGB thanks the ERC (Starting Grant No. 336679). NH is funded by the DFG-funded research unit INUIT (DFG-FOR-1525-6449 and MO668-4-2). AK is supported by the Helmholtz Association under Atmosphere and Climate Programme (ATMO). We acknowledge support by Prof. Dr. Thomas Leisner (IMKAAF, KIT) and useful comments from Dr. Yuki Nagata. We are grateful to continuous technical support of Mr. Marc-Jan van Zadel during the SFG measurements.

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
