# Peer review of "Surface charge-induced orientation of interfacial water suppresses heterogeneous ice nucleation on $\alpha$ -alumina (0001)"

_Atmospheric Chemistry and Physics, 2017_

## Referee Comment (RC1) · Anonymous Referee #3 · 25 Apr 2017

The authors shed light on some of the underlying mechanisms of heterogeneous nucleation of ice, and for that reason alone I am in favor of publication. I have outlined a few points below which the authors may wish to consider.

One substantial improvement to the paper would be to note the water activity for the various solutions used. While the emphasis is on pH and the resulting charge at the surface, noting $a_w$ would enable comparison with freezing point depressions as outlined in Koop et al. (2000). The statement on pg. 5, line 15 that the measured freezing temperature on Si wafers was the same for all pH solutions suggests that the water activity was very near 1 for all the solutions, but this should be confirmed.

[Figure]

The principal conclusion that I draw from this paper is that surfaces can overtemplate water. This has been known for a long time. The authors cite Fletcher's paper from 1959, where he shows that increasing the order of water molecule's in a pre-critical embryo too much can actually decrease the probability of freezing. Fletcher showed that there is an appreciable entropic penalty for nucleation on the basal plane of silver iodide because it is polar. In contrast, the penalty for nucleation on a prism face is negligible. The prism face of silver iodide still acts as a template for the ice embryo, but the degree of alignment for the water dipoles is mitigated because ions of both signs are exposed. For this reason, I think the conclusion (page 9, lines 7-8) which reads "Apparently, charge-induced surface templating is detrimental for ice nucleation, regardless of the sign of the surface charge." should be softened. That statement may be true for this system, over this range of conditions, but I do not think it is appropriate to state it generally. (I concede that when taken together with the sentence just before that, this applies to corundum. Perhaps just add "on this surface" right after "ice nucleation" to reinforce the point.)

The conclusions in this paper suggest that for any pH other than 7, that the critical ice embryo forms in the second or third layer of molecules away from the surface. If the water molecules right at the surface are too tightly bound and/or constricted to allow them to adopt the ice lattice, doesn't that imply that other water molecules are the ones actually forming the embryo? It seems most likely that it would be water molecules that were affected by the ordering imposed by the surface, but were perhaps still free to rotate and/or translate enough to adopt the ice lattice. Is there any indication of this in the data? (To be clear, I am not asking for an exhaustive re-analysis of the data. I am simply curious as to whether a signature like this could be gleaned from this data.)

Pg. 2, lines 13-14: "The real influence of temperature and supersaturation on the interaction between water molecules and dust particle surface has not yet been explored." This is an overstatement. This has been extensively investigated. We don't have a

definitive answer yet, but there are plenty of groups that have asked the question and contributed pieces to the puzzle.

**References**

Koop, T., Luo, B., Tsias, A., Peter, T., 2000. Water activity as the determinant for homogeneous ice nucleation in aqueous solutions. *Nature* **406**, 611–614. doi:10.1038/35020537

---

## Referee Comment (RC2) · Anonymous Referee #2 · 27 Apr 2017

Authors describe an experimental study where they studied immersion freezing of droplets that consist of known chemical compounds and are made of different pHs. Their conclusion is that droplet freezing temperature depends upon the pH and is more efficient when solution is neutral. Further, they use vibrational sum frequency generation spectroscopy technique to understand the structure of the water molecules at the interface of oxide material that is used to catalyze the nucleation. SFG results shows the sensitivity towards droplet freezing and intensity values (3400 and 3200 per cm). The results are novel, and they would help wider community to understand the more about the heterogeneous ice nucleation. I have the following comments, and after they are addressed, I recommend the manuscript for the publication.

General comments:

Introduction section needs text on atmospheric relevance of surface charge on the aerosol. How one can interpret/use these results in the context of understanding atmospheric ice formation. Some text describing how varying pH of droplets can be used to understand about the surface charge (interface between droplet and substrate) is required. Few details (OH group orientation etc) are already described on page 8.

Experimental: Describe rate of cooling and step size experiment. Not clear if it is cooled for one degree and hold T constant. Brief description of the cold-stage needs to be included. In section 3, it is mentioned that rate of cooling is 5 deg/min, this rate is different than described above. How droplets were visualized, using microscope? What is the typical size of droplet as you used different solutions or they remained constant, and can any images showing before and after freezing events be added in the main text. There is camera shown in figure 1. More details particularly magnification and model name are needed. How frozen fraction was calculated. How do you define a spectra?

Conclusion section is short and lacking many details. Describe what technique (because this is new about this work) was used, why alumina substrate was used and elaborate key results followed by the atmospheric implications. Remove any speculation and literature review text to add more clarity and improve readability.

Page 2, line 14-15: Needs reference

Page 2, Line 26-27: Elaborate and revise the sentence further.

Page 3, Line 1-3: Please add more specifics or details to understand why you want to take this particular approach.

Page 3, Line 12: Revise the sentence.

Page 4, Line 10-11: Revise the sentence.
Page 5, Line 11: pH 7.2 – is this water.

Page 5, Line 26: . . .investigated. . .

Page 8: Line 27. Can Fig 2c and d be included in Fig 4. These figure panels are not described until this point.

Minor comments: Peckhaus 2016a and b are similar.

Provide a space between references included within the text. Also a space is needed after 'period' located at the end of the sentence. See page 3, Line 10, 11.

Supplement section is incomplete.

---

## Referee Comment (RC3) · Anonymous Referee #1 · 28 Apr 2017

Abdelmonem et al. present a laboratory work on the effects of surface charge on ice nucleation at the interface between aqueous solution and alumina. Droplet freezing assay was used to determine the immersion freezing temperature of aqueous solutions with different pHs whereas the SFG spectroscopy was applied to investigate the interfacial water structures during the freezing processes on the same surface. With SFG information, this study provides some insights on the underlying ice nucleation mechanisms at molecular level. This will be a useful complementary technique additional to the freezing experiments. I recommend it for publication after considering the following comments in the revisions.

Comments:

[Figure]

1. In the Introduction section, it is suggested to have a briefly introduction on the SFG spectroscopy for readers who are not experts.

2. P1, L25-27, this statement is not correct as the heterogeneous ice nucleation could also occur below -38 degree C.

3. P1, L30-32, the sentence is overstated. It is not conclusive yet on the existence of actual active sites that induce ice nucleation.

4. P2, L2, Peackhaus et al 2016a and 2016b are in ACPD and ACP, respectively, final form should be cited?

5. P3, L15, how $\alpha$-alumina (0001) substrate was selected or treated before the freezing experiments, from which manufacturer?

6. P3, L38, and Figure 1, in the SFG setup, what is size of the sample area where the signal was collected, since the water drop in these experiments were large and freezing of ice could be initiated at the interface outside the sampling area. In addition, the $\alpha$-alumina (0001) should be homogenous and well cleaned.

7. P4, L20, what is "neat sapphire-c"?

8. P4, L24-26, please provide more information to support this statement. What are the pH values at the temperature close to the point of freezing? P6, L10, what are the uncertainties for the pH of solutions used in SFG experiments?

9. P8, L4-5, more detail description is needed for better understanding the use of this variable.

10. P8, L15, it is not clear what does "OH groups pointing with its H down to the bulk solution" mean? the OH groups near interface pointing away from the interface and into bulk solution?

11. P8, for the most part of the discussion on surface charge or surface, it is not always clear referring to which surface or which side of the interface or water molecular layer.

[Figure]

It will be informative and easy to understand these discussions if authors can provide an illustration.

12. P9, L7-8, apparently, this sentence is over stated since it is possible that the conclusion is only applied for certain types of particles/surfaces and pHs.

---

## Author Comment (AC1) · 19 May 2017

**Response to Referee #3**

*We thank the reviewer for careful reading the manuscript and for the positive comments and suggestions. We have changed the manuscript accordingly. Please find below a detailed response.*

RC: One substantial improvement to the paper would be to note the water activity for the various solutions used. While the emphasis is on pH and the resulting charge at the surface, noting aw would enable comparison with freezing point depressions as outlined in Koop et al. (2000). The statement on pg. 5, line 15 that the measured freezing temperature on Si wafers was the same for all pH solutions suggests that the water activity was very near 1 for all the solutions, but this should be confirmed.

*AC: The median freezing temperature measurements for the pH solutions on a Si substrate could be used to estimate the water activity in the solutions. Applying Hildebrand and Scott equation (Miyawaki et al., 1997) and assuming temperature measurements accuracy of ± 0.2 K the water activity has been calculated to be within the range of 0.993 to 0.996. This confirms our statement that the solute effect on variation of the freezing temperature is negligible.*

*This information will be added to the "Results and Discussion" text of the revised manuscript.*

**NOTE: The modified text will be posted in a separate "Author Comment". This will be the revised manuscript with tracked changes upon comments from all referees.**

RC: The principal conclusion that I draw from this paper is that surfaces can overtemplate water. This has been known for a long time. The authors cite Fletcher's paper from 1959, where he shows that increasing the order of water molecule's in a pre-critical embryo too much can actually decrease the probability of freezing. Fletcher showed that there is an appreciable entropic penalty for nucleation on the basal plane of silver iodide because it is polar. In contrast, the penalty for nucleation on a prism face is negligible. The prism face of silver iodide still acts as a template for the ice embryo, but the degree of alignment for the water dipoles is mitigated because ions of both signs are exposed. For this reason, I think the conclusion (page 9, lines 7-8) which reads "Apparently, charge-induced surface templating is detrimental for ice nucleation, regardless of the sign of the surface charge." should be softened. That statement may be true for this system, over this range of conditions, but I do not think it is appropriate to state it generally. (I concede that when taken together with the sentence just before that, this applies to corundum. Perhaps just add "on this surface" right after "ice nucleation" to reinforce the point.)

*AC: We agree with the referee. This conclusion was overstated. The text is now softened to ".... charge-induced surface templating is detrimental for ice nucleation on this particular surface, regardless of the sign of the surface charge."*

RC: The conclusions in this paper suggest that for any pH other than 7, that the critical ice embryo forms in the second or third layer of molecules away from the surface. If the water molecules right at the surface are too tightly bound and/or constricted to allow them to adopt the ice lattice, doesn't that imply that other water molecules are the ones actually forming the embryo? It seems most likely

that it would be water molecules that were affected by the ordering imposed by the surface, but were perhaps still free to rotate and/or translate enough to adopt the ice lattice. Is there any indication of this in the data? (To be clear, I am not asking for an exhaustive re-analysis of the data. I am simply curious as to whether a signature like this could be gleaned from this data.)

*AC: The reviewer raises a very interesting point. It is indeed not unlikely that the relatively mild templating occurring in water layers further from the surface (as evident from atomic force microscopy studies on similar systems) may be responsible for nucleation. Unfortunately, SFG does not allow to distinguish between contributions from different distances from the surface, so there is no reliable statement that we could make in this context.*

RC: Pg. 2, lines 13-14: "The real influence of temperature and supersaturation on the interaction between water molecules and dust particle surface has not yet been explored." This is an overstatement. This has been extensively investigated. We don't have a definitive answer yet, but there are plenty of groups that have asked the question and contributed pieces to the puzzle.

*AC: We totally agree that this sentence was an overstatement. It is replaced now by "The real influence of temperature and supersaturation on the interaction between water molecules and dust particle surface has been the focus of research since decades but remains debated."*

*Miyawaki, O., Saito, A., Matsuo, T., and Nakamura, K.: Activity and Activity Coefficient of Water in Aqueous Solutions and Their Relationships with Solution Structure Parameters, Bioscience, Biotechnology, and Biochemistry, 61, 466-469, doi: 10.1271/bbb.61.466, 1997.*

---

## Author Comment (AC2) · 19 May 2017

**Response to Referee #2**

*We thank the reviewer for careful reading the manuscript and for the positive comments and valuable recommendations. We have changed the manuscript accordingly. Please find below a detailed response.*

***NOTE: The modified manuscript text will be posted in a separate "Author Comment". This will be the revised manuscript with tracked changes upon comments from all referees.***

RC: Introduction section needs text on atmospheric relevance of surface charge on the aerosol. How one can interpret/use these results in the context of understanding atmospheric ice formation. Some text describing how varying pH of droplets can be used to understand about the surface charge (interface between droplet and substrate) is required. Few details (OH group orientation etc) are already described on page 8.

*AC: The surface charge of a metal oxide surface is pH dependent. The higher the pH with respect to the point of zero charge (pzc) of the surface the more negative is the surface and vice versa (Kosmulski, 2001, 2009). This process is controlled by the degree of protonation and deprotonation of the surface and the adsorption of dissolved ions. It is has been shown that the acidic particulates (e.g. $SO_4$, and $NO_3$) from anthropogenic sources decrease the pH values in cloud and rain water (Castillo et al., 1983; Scott, 1978). On the other hand, it was reported that alkaline particulates can be observed in regions where soil is rich with alkaline components, e.g. Ca and Mg, (Khemani et al., 1985b). In these regions high pH values were observed in cloud and rain water (Khemani et al., 1985a; Khemani et al., 1987).*

*Since immersion freezing is based on aerosol particles immersed in water droplets in a cloud for a certain time, we expect that their surface charge may change due to the variation in the droplet pH. Since surface charge is one of the surface properties which influence its interaction with water molecules, we believe that such results will help in understanding one of the not well explored parameters of ice nucleation in the atmosphere.*

*We agree with the referee, this part was not clear in the text. We have attempted to remedy this by including a discussion in the Introduction.*

RC: Experimental: Describe rate of cooling and step size experiment. Not clear if it is cooled for one degree and hold T constant. Brief description of the cold-stage needs to be included. In section 3, it is mentioned that rate of cooling is 5 deg/min, this rate is different than described above. How droplets were visualized, using microscope? What is the typical size of droplet as you used different solutions or they remained constant, and can any images showing before and after freezing events be added in the main text. There is camera shown in figure 1. More details particularly magnification and model name are needed. How frozen fraction was calculated. How do you define a spectra?

*AC: We thank the referee for pointing that the description of the cooling SFG experiment was not sufficient. The sample was cooled by one degree, e.g. from T to T-1 °C, with a rate of 1°C/min and hold temperature constant for one and half minute. Afterwards, a spectrum was collected. Each spectrum took 30 sec integration time. The sapphire prism was placed in a copper adaptor which was fixed on the silver block of the Linkam cold-stage. Detailed description and drawing of the assembly of*

*the SFG measuring cell can be found in (Abdelmonem, 2017). The cold-stage can perform controlled heating and cooling ramps, applied to the silver block, at rates between 0.01 and 100 °C/min. Temperature stability of the cold stage is better than 0.1 K.*

***NOTE:*** *We have used two cold-stages, one for the supercooled SFG experiments and one for the "cold-stage" experiments. We used the term "cold-stage experiments" for those of the freezing assay. This may be misleading and let the reader confuse what we are talking about. So, in the new version of the manuscript, we have referred to the "cold-stage experiment" by "droplet freezing assay setup"*

*We have changed the text in the manuscript accordingly.*

*In section 3, the cooling rate, of 5 °C/min, was that of the cold-stage experiments (not the SFG experiments). The cooling rate doesn't change the results, however, since the acquisition rate of the cold-stage experiments is 8 frames per second, i.e. much faster than that of the SFG, so that we could use higher cooling rate.*

*As mentioned in the manuscript, the typical droplet size in the SFG experiments was roughly 15µl and difficult to keep constant from experiment to experiment. The typical droplet size in the cold-stage experiments was (0.21 ±0.07) nL and kept constant for all experiments and solutions. This is why we rely on the cold-stage experiments to determine the median freezing temperature. We have added new panels to Fig. 1 and Fig.2 with images showing droplets before and after freezing events for SFG and cold-stage experiments respectively.*

[Figure]

*Figure 1. Upper panel: Schematic of the sample setup used in the SFG experiments. The SFG signal is generated in a co-propagating total internal reflection geometry at the spatial and temporal overlap of the incoming visible (800nm) and infrared (BB-IR) light, focused down to a ~100 μm diameter spot size. The camera (Guppy F-036 Allied Vision Technology with LINOS Macro-CCD Lens 0.14x (1:7) f4) is used to observe the droplet while placing it on the surface and filling the cell with oil, and to observe its status during the experiment. Lower panel: images of a typical drop on the substrate before and after freezing. Lower panel right: Typical water spectra before and after the freezing.*

[Figure]

*Figure 2. (a) Fraction of frozen droplets as a function of temperature of Sapphire (0001) surface in contact with different pH solutions. The black curve shows the data for water on a silicon substrate. (b) Freezing temperature as a function of pH of the different solutions for three independent experiments. **(c)** Images of typical droplets on the substrate at different stages of the droplet freezing assay experiment. (d) SFG spectra of liquid water at the Sapphire (0001) surface, right before the nucleation event (see text for details); for fits, see Fig. 3. (e) Absolute value of the amplitude divided by width as a function of pH for the 3200 cm$^{-1}$ resonance inferred from the data shown in (d). Different data points indicate data obtained from freshly prepared surfaces, recorded on different runs.*

*Frozen fraction, in the cold stage experiments, is defined as the number of frozen droplets to the total number of droplets printed on the surface. In the SFG experiments, we probe only one droplet per run and the spectra at the solution water interface are collected as a function of temperature as described above.*

*The camera in figure 1 is Guppy F-036 Allied Vision Technology with LINOS Macro-CCD Lens 0.14x f4 and magnification (1:7). Specifications have been added to the caption of Fig.1*

RC: Conclusion section is short and lacking many details. Describe what technique (because this is new about this work) was used, why alumina substrate was used and elaborate key results followed by the atmospheric implications. Remove any speculation and literature review text to add more clarity and improve readability.

*AC: Indeed the conclusion was much shortened and assumed that the reader is familiar with the technique. The conclusion text has been expanded significantly to:*

[revised manuscript text omitted]

*This sentence was deleted after revising the former one (Page 3, Line 1-3 in the original manuscript)*

RC: Page 4, Line 10-11: Revise the sentence.

*AC:*

*Original sentence:*

*"The heterogeneous nucleation temperatures mentioned in the results and discussion section are those obtained from the cold-stage study."*

*Revised sentence:*

*"The heterogeneous nucleation temperatures reported in the "Results and Discussion" section were obtained from the cold-stage results shown, e.g., in Fig. 2a "*

RC: Page 5, Line 11: pH 7.2 – is this water.

*AC: Yes*

Page 5, Line 26: : : :investigated: : :

*AC: Corrected*

RC: Page 8: Line 27. Can Fig 2c and d be included in Fig 4. These figure panels are not described until this point.

*AC: One of the main points of the manuscript is the anti-correlation between median freezing temperature and SFG intensity; it is for this reason that we think it is important that panels 2b and 2d appear in one figure.*

RC: Minor comments: Peckhaus 2016a and b are similar.

*AC: Corrected*

RC: Provide a space between references included within the text. Also a space is needed after 'period' located at the end of the sentence. See page 3, Line 10, 11.

*AC: Done*

RC: Supplement section is incomplete.

*AC: We are not quite sure what the reviewer is referring to…*

---

## Author Comment (AC3) · 19 May 2017

**Response to Referee #1**

*We thank the reviewer for the careful reading of our manuscript and for the valuable suggestions. We have changed the manuscript accordingly. Please find below a detailed response.*

RC: 1. In the Introduction section, it is suggested to have a briefly introduction on the SFG spectroscopy for readers who are not experts.

*AC: Thanks to the referee to point out the necessity of considering those readers who are not expert in SFG. We have included, in the introduction, a brief general introduction to the technique. In addition, we have expanded the description of the experimental setup in the "Experimental" section. These changes in addition to the notes in the SI will help the readers with different backgrounds to understand the usefulness of the technique.*

***NOTE: The modified text will be posted in a separate "Author Comment". This will be the revised manuscript with tracked changes upon comments from all referees.***

RC: 2. P1, L25-27, this statement is not correct as the heterogeneous ice nucleation could also occur below -38 degree C.

*AC: we have changed the phrasing accordingly, and now state that "homogeneous nucleation occurs typically at temperatures below -38° C".*

RC: 3. P1, L30-32, the sentence is overstated. It is not conclusive yet on the existence of actual active sites that induce ice nucleation.

*AC: In some special cases (K-rich feldspars, ice nucleating bacteria) the evidence is conclusive. Nevertheless, we have tempered the sentence.*

*Original sentence: "Ice nucleation occurs preferentially at microscopic active sites and is dominated by the nature of these sites rather than by the average behavior of the surface"*

*Revised sentence: "It is suggested that ice nucleation occurs preferentially at microscopic active sites and is dominated by the nature of these sites rather than by the average behavior of the surface"*

RC: 4. P2, L2, Peackhaus et al 2016a and 2016b are in ACPD and ACP, respectively, final form should be cited?

*AC: We thank the reviewer for pointing out this omission. We have corrected this oversight in the revised version.*

RC: 5. P3, L15, how _-alumina (0001) substrate was selected or treated before the freezing experiments, from which manufacturer?

*AC: This information was provided on P4, L18-26 in the original manuscript.*

6. P3, L38, and Figure 1, in the SFG setup, what is size of the sample area where the signal was collected, since the water drop in these experiments were large and freezing of ice could be initiated at the interface outside the sampling area. In addition, the _-alumina (0001) should be homogenous and well cleaned.

*AC: The area sampled by the SFG beams is approximately 100 μm in diameter. It is indeed possible that nucleation occurs outside the sampled area, but since nucleation and growth across the droplet occur on a timescale short compared to the measurement time, this will not affect the results. We have included the information on the beam spot size in the caption of figure 1. The α-alumina (0001) was homogenous and well cleaned (details are giving in the experimental section, P4, L18-26 in the original manuscript).*

7. P4, L20, what is "neat sapphire-c"?

*AC: It is the α-alumina (0001) cut. We probably should not have used double definitions. This has been corrected in the revised version.*

RC: 8. P4, L24-26, please provide more information to support this statement. What are the pH values at the temperature close to the point of freezing? P6, L10, what are the uncertainties for the pH of solutions used in SFG experiments?

*AC: We assume that the referee means the sentence P5, L4-6 in the original manuscript where we speak about change of pH with cooling. It is hard to estimate the pH values at the freezing temperature. However, as stated in the link below, "the change in solution pH with temperature is controlled by the temperature-dependent dissociation constants. The effect of temperature on solution pH increases as the pH approaches the dissociation constant. For strong acids and bases, the dissociation constants are beyond the 0 to 14 pH range, so the primary effect is the water dissociation constant that is $1 \times 10^{-14}$ at 25°C. Consequently, the effect of temperature is substantial for basic solutions" (see figure below).*

[Figure]

"

https://www.isa.org/standards-and-publications/isa-publications/intech-magazine/2009/december/web-exclusive-opportunities-for-smart-wireless-ph-conductivity-measurements/

*In any case, the order of acidity strength of our solutions at room temperature doesn't change at lower temperatures. For the experiments, what matters in the end, is the effect of pH on the surface charge. Since we measure the effect of surface charge on water structuring using SFG spectroscopy, the variation of the actual pH with temperature is not critical for the conclusions drawn from our work. We have included a brief discussion in the revised manuscript at the end of the Experimental section. We write:*

*"In addition, the pH is only tuned to change the surface charge, which is determined independently from the water alignment using SFG. Hence, knowledge of the precise temperature dependence of the pH is not required to correlate surface charge to freezing temperature."*

RC: 9. P8, L4-5, more detail description is needed for better understanding the use of this variable.

*AC: This variable represents the strength of the oscillator normalized to the linewidth of the band, which provides a direct measure for the number density of the different surface groups. The sentence is revised in the new version of the manuscript.*

*Revised sentence: "Figure 4 depicts the amplitude divided by the width of each resonance ($\vec{A}_q/2\Gamma_q$). This variable represents the strength of the oscillator normalized to the homogeneous broadening of the band and thus is a measure for the number density of the different surface groups (Backus et al., 2012a; Li et al., 2015; Pranzetti et al., 2014)."*

*In addition, more details on the fitting variables are given now in the Introduction.*

RC: 10. P8, L15, it is not clear what does "OH groups pointing with its H down to the bulk solution" mean? the OH groups near interface pointing away from the interface and into bulk solution?

*AC: Yes. The word "down" is now replaced with "into" in the revised version*

RC: 11. P8, for the most part of the discussion on surface charge or surface, it is not always clear referring to which surface or which side of the interface or water molecular layer. It will be informative and easy to understand these discussions if authors can provide an illustration.

*AC: We thank the referee for this practical suggestion. In the revised version, we have included an inset in figures 3a, b, c and d to illustrate the net dipole orientations of water on the surface at different pHs (surface charges).*

[Figure]

*Figure 3. Measured (points) and fitted (lines) SFG spectra of α-Al2O3 (0001)/water interfaces collected in SSP polarization during the cooling cycles of (a) pH 3 - HNO3, (b) pH 7, (c) pH 9 - NaOH and (d) pH 9 - NH4OH. The spectra shown here are always the last and first spectrum right before (red) and immediately after (blue) the freezing event, respectively. The signs of the resonances are indicated by a circle with + or - sign (see SI for details). The insets in a, b, c and d illustrate the net dipole orientations (from O to H, as inferred from the low frequency peak) of water on the surface at different pHs (surface charges).*

12. P9, L7-8, apparently, this sentence is over stated since it is possible that the conclusion is only applied for certain types of particles/surfaces and pHs.

*AC: That's correct. We changed the sentence to "… charge-induced surface templating is detrimental for ice nucleation on this particular surface, regardless of the sign of the surface charge"*

---

## Author Comment (AC4) · 19 May 2017

The authors are very grateful to the careful review and valuable recommendations from all referees which certainly improved the quality of the manuscript. The individual rebuttals were submitted in an earlier time. Here we attach the revised manuscript, with tracked changes, after all suggestions and recommendations from the three referees.

Please also note the supplement to this comment:
http://www.atmos-chem-phys-discuss.net/acp-2017-224/acp-2017-224-AC4-supplement.pdf

[Figure]

[Figure]

**Supplement:**

[revised manuscript text omitted]